# The Corrosion Inhibition of Montmorillonite Nanoclay for Steel in Acidic Solution

**DOI:** 10.3390/ma16186291

**Published:** 2023-09-20

**Authors:** Ehab AlShamaileh, Abdelmnim M. Altwaiq, Ahmed Al-Mobydeen, Imad Hamadneh, Bety S. Al-Saqarat, Arwa Hamaideh, Iessa Sabbe Moosa

**Affiliations:** 1Department of Chemistry, School of Science, The University of Jordan, Amman 11942, Jordan; i.hamadneh@ju.edu.jo (I.H.); dr.iessa89@gmail.com (I.S.M.); 2Department of Chemistry, College of Arts and Sciences, University of Petra, Amman 11196, Jordan; aaltweiq@uop.edu.jo; 3Department of Chemistry, Faculty of Science, Jerash University, Jerash 26150, Jordan; ahmeddd_mob@yahoo.com; 4Department of Geology, The University of Jordan, Amman 11942, Jordan; 5Water, Energy and Environment Research and Study Center, The University of Jordan, Amman 11982, Jordan; arwa.efb@gmail.com

**Keywords:** steel, corrosion inhibition, montmorillonite nanoclay, acidic solution

## Abstract

The aim of this research is to study the anticorrosive behavior of a coating consisting of modified montmorillonite nanoclay as an inorganic green inhibitor. The anticorrosion protection for mild steel in 1.0 M HCl solution is studied via weight loss, electrochemical methods, SEM, and XRD. The results proved that montmorillonite nanoclay acts as a good inhibitor with a mixed-type character for steel in an acidic solution. Both anodic and cathodic processes on the metal surface are slowed down. There is a clear direct correlation between the added amount of montmorillonite nanoclay and the inhibition efficiency, reaching a value of 75%. The inhibition mechanism involves the adsorption of the montmorillonite nanoclay onto the metal surface. Weight loss experiments are carried out with steel samples in 1.0 M HCl solution at room temperature, and the same trend of inhibition is produced. SEM was used to image the surface at the different stages of the corrosion inhibition process, and also to examine the starting nanoclay and steel. XRD was used to characterize the nanoparticle structure of the coating. Montmorillonite nanoclay is an environmentally friendly material that improved the corrosion resistance of mild steel in an acidic medium.

## 1. Introduction

Mild steel is a common material used in many industrial and domestic applications, including construction, containers, pipelines, and structures. Mild steel is a type of carbon steel with a low amount of carbon (typically a wt% of 0.05% to 0.25%); also known as “low-carbon steel”. Corrosion is considered as the most challenging problem for steel, especially when in contact with acidic media [1]. Huge financial and ecological losses are incurred due to the corrosion of steel worldwide. Hydrochloric acid (HCl) and sulfuric acid (H_2_SO_4_) are the two most commonly used acidic solutions for cleaning industrial equipment [2]. Anti-corrosion materials (corrosion inhibitors) are usually used in the process, for the protection of metals [1,3,4,5,6]. However, these materials are harmful and expensive, due to their chemical nature. Organic corrosion inhibitors are widely used in steel protection, but they suffer from instability and toxicity [7]. Natural and green corrosion inhibitors are also used in the protection of steel [8]. A high amount of research is being undertaken in the field of finding the optimum corrosion inhibitor for various industrial applications, especially in acidic media. Mohamed et al. studied the inhibition of zinc of hybrid Ti/Ce salt with various concentrations, using weight-loss and potentiodynamic polarization in 1M HCl solution [9]. Researchers have investigated the corrosion inhibition property of sodium lignosulfonate in HCl, using three different independent monitoring techniques, which are weight-loss measurements, electrical conductance, and potentiodynamic polarization [10,11]. Many researchers have studied the corrosion inhibition of steel and other metals in various environments, including HCl [4,12,13,14,15,16,17,18,19,20,21,22]. The mechanism of inhibition depends on many factors, such as the metal type, medium type and concentration, nature of the inhibitor, and temperature [23]. Natural clays are materials that are economically affordable, with excellent mechanical properties and almost zero toxicity levels. Montmorillonite is a type of silicate mineral with a typical TOT layered structure (tetrahedral–octahedral–tetrahedral or, in other words, an octahedral layer sandwiched between two tetrahedral layers), with an interlaminar domain in layers of TOT that is excellent for adsorption and other properties. A review of the recent advances in the organic modification of montmorillonite/alkylammonium compound processes, the adsorption characteristics of different modifiers on montmorillonite, the structure characteristics and gel performance of montmorillonite/alkylammonium, and the influence mechanism, is published [24]. Researchers investigated the corrosion-protection behavior of natural montmorillonite clay as a coating, and found that the addition of clay nanolayers improved the corrosion resistance of the coating [25]. Farahi et al. described a new approach to modifying a coating formulation, by adding sodium montmorillonite, to increase the corrosion resistance properties for carbon steel in HCl [26]. Messinese et al. have published a comprehensive investigation on the effects of surface finishing on the resistance of stainless steel to localized corrosion in chloride-rich environments [27]. Tambovskiy et al. described a method for protecting low-carbon steel using potentiodynamic polarization curves in a chloride solution [28]. Howyan et al. have investigated the protection efficiency of carbon steel using a clay-based coating, and achieved a value of 81.4% via electrochemical measurements [29]. The present study employs the use of the modified natural clay montmorillonite in the nano form (MMT) as a coating for the protection of a mild steel surface in an acidic medium (1.0 M HCl). The anticorrosion protection for mild steel in 1.0 M HCl solution is studied via weight loss and electrochemical methods (potentiodynamic polarization). Scanning electron microscopy (SEM) is used to characterize the tested surfaces. Moreover, XRD is used to study the structure of the montmorillonite nanoclay. The literature is abundant with research and reviews stressing the importance and viability of clay-based coatings across diverse engineering materials [30,31,32,33].

## 2. Materials and Methods

### 2.1. Weight-Loss Measurements

The modified montmorillonite (MMT) used in this study was purchased from Sigma-Aldrich, Darmstadt, Germany (montmorillonite nanoclay, surface modified, containing 35–45 wt% dimethyl dialkyl (C14-C18) amine, SKU: 682624-500G, particle size: ≤20 μm). All other chemicals (NaOH and HCl) were purchased from local providers, with the highest possible purity (analytical grade). Carbon steel sheets were purchased locally (JOST, Jordan Steel Group, Amman, Jordan), and cut to the size of 2 cm × 2 cm × 0.2 for weight-loss measurements. All masses were measured using a digital micro-balance (Model SEJ205, Taipei, Taiwan). For the potentiodynamic polarization studies, the exposed area of the steel surface (working electrode) was adjusted to be around 4 cm^2^. The electrodes were ground with emery papers of a fine grade (800), and degreased with acetone, and rinsed with distilled water before each experiment. Distilled water was used to prepare all solutions of 1.0 M HCl. For the weight-loss measurements, steel samples were ground, cleaned, dried in a stream of nitrogen, and then weighed accurately (using a 5-digit analytical balance). Each of the samples was then fully immersed in a 50 mL beaker, containing 1 M HCl and a different concentration of the inhibitor, MMT. The solutions were kept in a water bath set at room temperature. The steel samples were removed after 24 h and 72 h of immersion, washed with distilled water, dried, and weighed. At least three steel samples were used to produce an average value for the weight loss.

### 2.2. Electrochemical Measurements

Electrochemical experiments were carried out with a VoltaLab PGZ 100 potentiostat (Radiometer Analytical, Villeurbanne, France) in a double-wall three-electrode glass cell. Prior to the measurements, the surface of the working electrode (carbon) was carefully polished with alumina slurry, rinsed several times with distilled water, and then sonicated for about 1 min. All potential values are reported versus the saturated calomel electrode (SCE) as a reference electrode, and all measurements were carried out at room temperature. A platinum wire was used as the auxiliary electrode. All glassware for electrochemical experiments was carefully cleaned via immersion in a solution of concentrated sulfuric acid containing ammonium peroxydisulfate (NH_4_)_2_S_2_O_8_ overnight, followed by ample rinsing with distilled water. Carbon samples were immersed in the test solution for a few minutes, until a steady open-circuit potential (OCP) was attained. The polarization curves were measured from a cathodic potential of −100 mV to an anodic potential of +100 mV, with respect to the open circuit potential (OCP) at a scan rate of 10 mV/s. The generated Tafel plots were analyzed via extrapolation, to evaluate the corrosion potential (Ecorr) and the corrosion current densities (Icorr). Several measurements were carried out for each experiment, to ensure the reproducibility of the data.

### 2.3. XRD Measurements

XRD analysis was employed using (Malvern Panalytical Aeris, Cu k_α1_, 0.15406 nm 0.01 step angle, Almelo, The Netherlands) to calculate the d-spacing values, which, in turn, give an idea about the degree of modification in the montmorillonite nanoclay. The analysis of the clay was performed via comparison of the peak positions and intensities to those on spectra from the XRD diffraction data library. The diffraction patterns were measured in the range of 5–80° 2θ for the raw and modified montmorillonite samples.

### 2.4. SEM Investigation

SEM examinations were conducted, to study the starting carbon steel and the surface morphology of steel samples, before and after protection with the montmorillonite nanoclay. The SEM (Inspect F50-FEI Company, Eindhoven, The Netherlands) was used for microstructure investigation and chemical analysis, via its integrated energy dispersive spectrometer (EDS). The prepared specimens were carefully washed with distilled water and alcohol, and dried, prior to the SEM investigation, after being fixed onto aluminum stubs with a double-adhesive carbon sticker. A specimen of as-received carbon steel with an area of about 1 cm^2^ was ground using different emery paper grades of 200–1200, with good washing between each step of grinding. The specimen was polished using diamond pastes of 7 µm, 3 µm, and 0.25 µm, and then etched with a 3% nital solution (3% nitric acid) for microstructure revealing in the SEM. The starting material (montmorillonite nanoclay) was also examined, to explore the particle size range, chemical analysis, and the shape of particles, in general.

## 3. Results and Discussion

### 3.1. Weight-Loss Study

Table 1 shows the results of the weight-loss experiments. It is evident that the modified montmorillonite nanoclay (MMT) inhibits the corrosion rate of steel in 1.0 M HCl, and the inhibition efficiency increases with an increase in the inhibitor concentration. We can see that the best inhibition efficiency was achieved around 9.00 × 10^−2^ gMMT/mL HCL. This may be attributed to the adsorption of the inhibitor onto the metal surface. The inhibition efficiency is defined as the percentage of corrosion decrease upon the addition of an inhibitor, compared to that without the inhibitor. The effect of varying the immersion time and temperature of the solution on the inhibition efficiency was studied for selected concentrations, but no apparent effect was noticed. However, all experiments were carried out at 25 °C, using a water bath, for 24 h. We notice that, as the concentration of MMT increases, the inhibition efficiency increases, with a maximum of 73.7% at the highest concentration (9.00 × 10^−2^ gMMT/mL HCl). We decided not to go further, as we noticed that the solubility of MMT becomes less and less with the increase in concentration. Upon repeating the experiment from the beginning, but this time for a longer period of time (72 h), we noticed the same trend in the protection. The gained maximum inhibition efficiency of 79.9% shows that the system reached equilibrium in less than 72 h (Table 2), and that there is no need for a longer immersion time over 24 h. Moreover, this extra gained protection of around 8% does not justify the extra two days of excessive immersion. The corrosion rate (in mg/h·cm^2^) is calculated via dividing the mass change (in mg) by the sample surface area and the time in hours. The inhibition efficiency is calculated via dividing the difference in corrosion rate (between the sample and the blank) by the corrosion rate of the blank.

### 3.2. Electrochemical Study

The potentiodynamic polarization curves of steel in 1.0 M HCl solution, without and with different concentrations of the inhibitor MMT, after an immersion time of 24 h, are shown in Figure 1. It is observed that both the cathodic and anodic processes are hindered upon the addition of MMT. The polarization curves were recorded for inhibitor concentrations ranging between 1.25 × 10^−2^ g MMT/mL HCl and 9.0 × 10^−2^ g MMT/mL HCl, at room temperature. The best % inhibition efficiency of 72% was reached at the highest concentration of MMT. The electrochemical parameters, such as corrosion current density (Icorr), corrosion potential (Ecorr), Tafel constants, βa and βc, and % inhibition efficiency were calculated from Tafel plots (Table 3). It is observed that the presence of the inhibitor lowers the corrosion current density, which reaches a minimum value at the highest inhibitor concentration. It is also observed that the Ecorr values and the Tafel constants βa and βc did not change significantly as a function of the inhibitor concentration, indicating that MMT behaves as a mixed-type inhibitor. The same experiments were performed with an immersion time of 72 h, and the results are shown in Figure 2 and Table 4. Compared to the 24 h immersion time, the results showed a similar trend, and a slightly better % inhibition efficiency (75%). This may be attributed to reaching an equilibrium state of adsorption and, hence, inhibition, with a time of immersion of less than 72 h. Additionally, this indicates that there is not much need for an immersion time longer than 24 h. The corrosion rate is an output of the software (voltamaster 4) in units of mm/Y. The inhibition efficiency is calculated via dividing the change in corrosion rate (the sample–blank) by the corrosion rate of the blank. 

### 3.3. XRD Analysis

The XRD results revealed characteristics of the modified montmorillonite clay, due to diffraction peaks at d-spacing 4.3 Å and 3.3 Å, at 2θ = 20.82° and at 2θ = 26.64°, respectively (Figure 3). Upon the modification of the clay, the XRD patterns showed a new peak at 2θ = 6.81°, corresponding to an interlayer spacing of 13.0 Å. This noticeable expansion is a clear proof of modification within the layers. Moreover, the nanoclay showed a mixed structure of crystalline–amorphous in the angle range of 2θ = 40–85°, and that is probably due to the nanoscale particle size of the modified montmorillonite used.

### 3.4. SEM Study

Scanning electron microscopy, coupled with the energy dispersive spectrometer (SEM–EDS), was used to study the morphology and elemental composition of the starting mild steel surface, as well as the layer formed on the mild steel surfaces before and after exposure to the corrosive solution (1.0 M HCl), with and without the MMT coating, immersed for 72 h. Figure 4a shows the etched specimen microstructure of the steel used, with its EDS spectrum, from which it can be seen that the steel was almost pure iron, with a small amount of carbon. The carbon is concentrated in the dark spots that appear in the image of Figure 4a as an Fe-C compound with a mass% of iron and carbon of approximately 99.76 and 0.22, respectively, as confirmed via the EDS unit. Figure 4b,c show SEM images of the steel surfaces in 1.0 M HCl, before and after immersion for 72 h in a 0.09 g/mL concentration of MMT, respectively. It seems that the steel surface without MMT is severely corroded, with porous-like structures (Figure 4b).

As for the steel sample immersed for 72 h in a 0.09 g/mL concentration of MMT, we clearly observed a protective layer on the surface, with very little corrosion (Figure 4c). We noticed the presence of some MMT particles adsorbed onto the surface. The corresponding elemental analysis of each steel sample is shown in Figure 4, on the right side of each image. 

This carbon percentage is attributed to the inherent composition of the steel sample, where the largest percentage is of iron. In the presence of the MMT coating, silicon and aluminum are present in the sample, which confirms the formation of the MMT coating. The percentages of elements for each surface obtained from the EDS analysis after 72 h immersion in 1.0 M HCl are summarized in Table 5. It is clear that the carbon content increased heavily upon the addition of the MMT, and this is attributed to the original carbon content in the MMT used. Figure 5 displays the SEM micrographs (SEIs) of the MMT powder used as a corrosion inhibitor at three magnifications, 10,000×, 20,000×, and 40,000×. It is clear that the starting material is in the nanoscale range.

## 4. Conclusions

In this research, a coating consisting of modified montmorillonite nanoclay as a green anticorrosive material for steel was explored, which represents the novelty of the research idea. The results of all the techniques used; the weight-loss, electrochemical, SEM and conjugated EDS, and XRD methods; were all in line with the protective behavior. The SEM images and the obtained EDS spectra proved that the steel used was almost pure iron, with some inclusions of the Fe-C compound. Furthermore, the results showed that there is a clear direct correlation between the amount of montmorillonite nanoclay and the inhibition efficiency. The inhibition mechanism involves the adsorption of the montmorillonite nanoclay onto the metal surface. As it is an environmentally friendly and safe material, montmorillonite nanoclay is an excellent corrosion inhibitor for mild steel in any acidic medium. Novel ideas to control metallic corrosion are very significant steps in the industrial sector, to extend the life of used metals, and to reduce production and maintenance costs. For practical applications, more research and testing is needed, to compare the current corrosion inhibitor to already-existing inhibitors, as well as to test the resistance of the coatings to scrubbing during the different processing stages.

## Figures and Tables

**Figure 1 materials-16-06291-f001:**
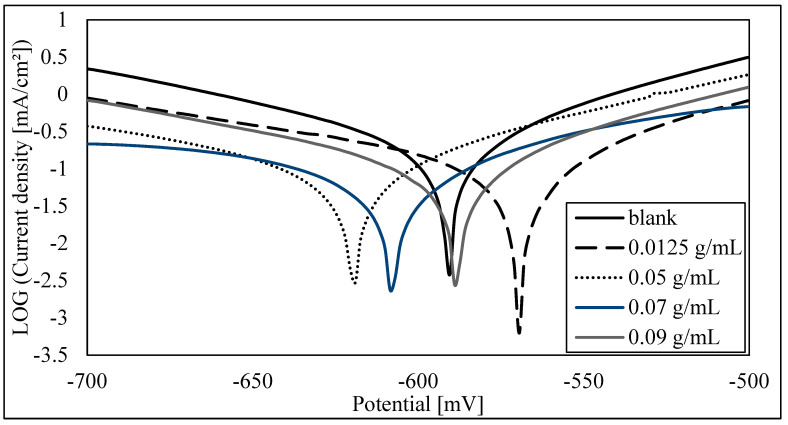
Polarization curves for the corrosion of steel in 1.0 M HCl, before and after immersion for 24 h in different concentrations of MMT. Scan rate: 100 mV/min.

**Figure 2 materials-16-06291-f002:**
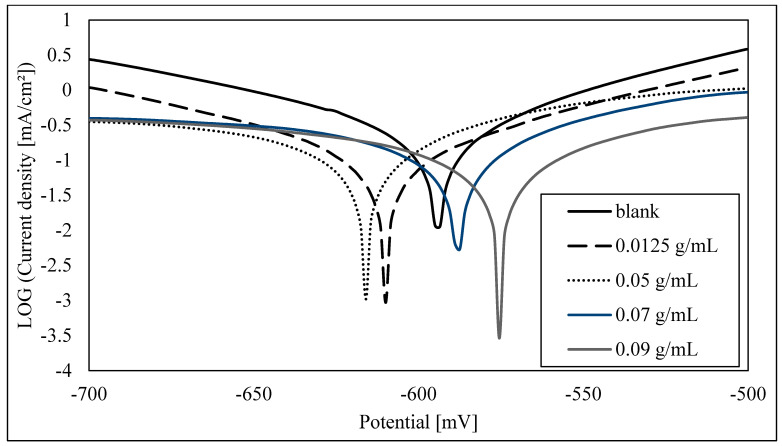
Polarization curves for the corrosion of steel in 1.0 M HCl, before and after immersion for 72 h in different concentrations of MMT. Scan rate: 100 mV/min.

**Figure 3 materials-16-06291-f003:**
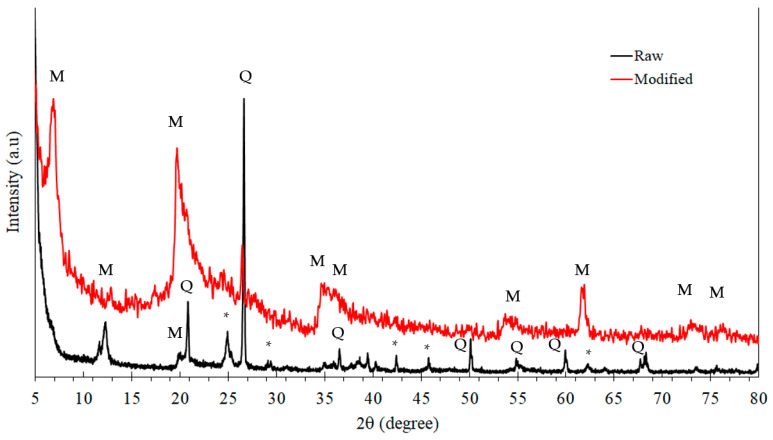
X-ray diffraction patterns of the raw montmorillonite clay, and the modified montmorillonite. M: montmorillonite, Q: quartz, *: illite, others.

**Figure 4 materials-16-06291-f004:**
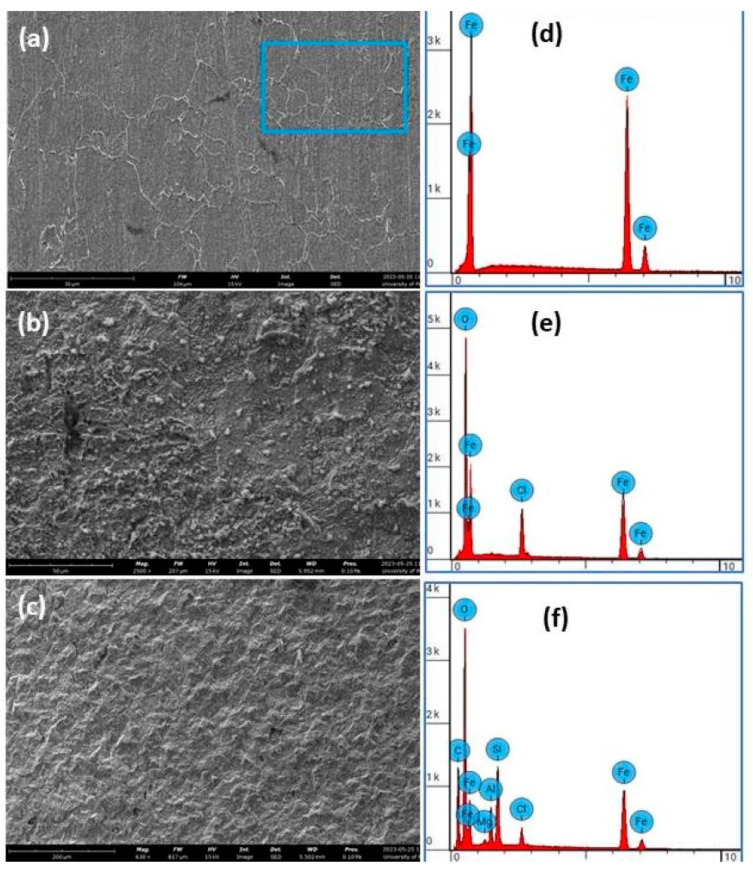
SEM images of mild steel surfaces with the corresponding EDS analysis spectra (on the right of each image). (**a**) The clean steel surface, (**b**) the surface after being immersed in 1.0 M HCl for 72 h, and (**c**) the surface coated with MMT, immersed in 1.0 M HCl for 72 h. (**d**) is the EDS for image (**a**), (**e**) is the EDS for image (**b**) and (**f**) is the EDS for image (**c**).

**Figure 5 materials-16-06291-f005:**
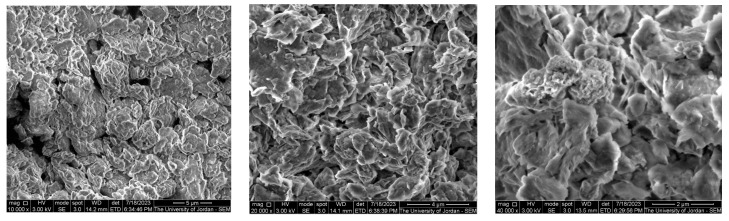
SEM micrographs (SEIs) of the MMT powder used as a corrosion inhibitor at three magnifications: 10,000×, 20,000×, and 40,000×.

**Table 1 materials-16-06291-t001:** The percentage inhibition efficiency of MMT from weight-loss experiments of steel in 1.0 M HCl. The time of immersion was 24 h for all samples. The surface area was approximately 4.3 cm^2^.

Sample No.	MMT Mass% g MMT/mL HCl	Mass before Immersion for 24 h (g)	Mass after Immersion for 24 h (g)	Mass Change g	Corrosion Rate (mg/h·cm^2^)	Inhibition Efficiency %
0 (blank)	0	0.7026	0.6452	0.0574	0.556	
1	1.25 × 10^−2^	0.7254	0.6840	0.0414	0.401	27.9
2	3.00 × 10^−2^	0.6987	0.6585	0.0402	0.390	30.0
3	5.00 × 10^−2^	0.7141	0.6754	0.0387	0.375	32.6
4	7.00 × 10^−2^	0.6666	0.6370	0.0296	0.287	48.4
5	9.00 × 10^−2^	0.6562	0.6411	0.0151	0.146	73.7

**Table 2 materials-16-06291-t002:** The percentage inhibition efficiency of MMT from weight-loss experiments of steel in 1.0 M HCl. The time of immersion was 72 h for all samples. The surface area was approximately 4.3 cm^2^.

Sample No.	MMT Mass% g MMT/mL HCl	Mass before Immersion for 72 h (g)	Mass after Immersion for 72 h (g)	Mass Change g	Corrosion Rate (mg/h·cm^2^)	Inhibition Efficiency %
0 (blank)	0	0.6485	0.5403	0.1082	0.349	
1	1.25 × 10^−2^	0.6658	0.5587	0.1071	0.346	1.0
2	3.00 × 10^−2^	0.6561	0.5547	0.1014	0.328	6.3
3	5.00 × 10^−2^	0.6544	0.5761	0.0783	0.253	27.6
4	7.00 × 10^−2^	0.7236	0.6801	0.0435	0.141	59.8
5	9.00 × 10^−2^	0.7211	0.6993	0.0218	0.070	79.9

**Table 3 materials-16-06291-t003:** Electrochemical corrosion parameters of steel in 1.0 M HCl, before and after immersion for 24 h in different concentrations of MMT. The atomic mass is 55.8 g/mol, the valence is 3, and the density is 7.9 g/cm^3^.

	Blank 0.5 M HCL	1.25 × 10^−2^ g MMt/mL HCl	5.00 × 10^−2^ g MMt/mL HCl	7.00 × 10^−2^ g MMt/mL HCl	9.00 × 10^−2^ g MMt/mL HCl
E (i = 0) (mV)	−595.7	−574	−624.4	−610.4	−593.3
Corrosion current (i_corr_) (µA/cm^2^)	158.2994	89.4206	70.381	60.4446	44.2554
Rp (ohm.cm^2^)	76.72	177.63	199.1	301.52	173.13
Beta anodic (mV)	60.2	67.2	70.7	81.3	41.1
Beta cathodic (mV)	−81.2	−117.9	−98.9	−117.8	−54.1
Corrosion rate (mm/Y)	1.202	0.6792	0.5345	0.4591	0.3361
Corrosion inhibition efficiency %	-	43%	56%	62%	72%

**Table 4 materials-16-06291-t004:** Electrochemical corrosion parameters of steel in 1.0 M HCl, before and after immersion for 72 h in different concentrations of MMT. The atomic mass is 55.8 g/mol, the valence is 3, and the density is 7.9 g/cm^3^.

	Blank 0.5 M HCL	1.25 × 10^−2^ g MMt/mL HCl	5.00 × 10^−2^ g MMt/mL HCl	7.00 × 10^−2^ g MMt/mL HCl	9.00 × 10^−2^ g MMt/mL HCl
E (i = 0) (mV)	−598.9	−614.5	−618.6	−590.2	−578.6
Corrosion current (i_corr_) (µA/cm^2^)	199	99.9354	80.6658	67.2648	49.3452
Rp (ohm.cm^2^)	62.13	133.59	125.05	126.82	187.51
Beta anodic (mV)	64	77.4	54.3	43.2	55.6
Beta cathodic (mV)	−78	−77.9	−76.3	−66.8	−63.4
Corrosion rate (mm/Y)	1.511	0.759	0.6127	0.5109	0.3748
Corrosion inhibition efficiency %		50%	59%	66%	75%

**Table 5 materials-16-06291-t005:** Percentage of elements for each surface, obtained from EDS analysis, after 72 h immersion in 1.0 M HCl.

Sample	Element (Weight%)					
	C	Cl	O	Si	Al	Fe
Steel, clean	0.22					99.76
Steel in HCl		10.54	25.10			64.36
MMT–steel in HCl	18.22	2.50	29.93	11.11	7.31	33.23

## Data Availability

Not applicable.

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
