# Peer review of "The Corrosion Inhibition of Montmorillonite Nanoclay for Steel in Acidic Solution"

_materials, 2023, doi:10.3390/ma16186291_

Round 1
Reviewer 1 Report
- The authors studied the effect of a modified MMT nanoclay as a corrosion inhibitor for steel in acid solution. The steel is immersed in a HCl solution with MMT. Since this inhibitor is supposed to compete with commercial inhibitors, it would have been useful if the authors had compared MMT with a couple of conventional inhibitors. This should be proposed for the next phase of the work. Also, it would have been useful to test the resistance of the coatings to scrubbing, as this may be needed during cleaning.
- Section 3.1: the authors should define “inhibition efficiency”.
- Also in section 3.1 the authors write “We decided not to go further as we noticed that the solubility of MMT becomes less and less with concentration.” What is the solubility of MMT in acid solution ? Does it precipitate at the highest concentrations ?
- Regarding Table 3, the authors should clarify the meaning of the following parameters: smoothing, calculation zone, segment and coefficient. They should also explain the influence of these parameters in the electrochemical calculations ?
- The authors write that they used “montmorillonite nanoclay, surface modified with 35-45 wt. % di- methyl dialkyl (C14-C18) amine”. They should clarify why they chose this form of MMT and not the plain MMT and what kind of role the organic part played in the corrosion protection. Is ist supposed that the surface become hydrophobic ? If it is the case contact angle measurements should be made. What kind of bonding exists between the steel and the MMT ? Is the bond formed with the organic or inorganic part of MMT ?
- In section 3.4 the authors perform the characterization of the MMT powder. Since this is a raw material that has been used all along the study, its characterization should have been performed at the beginning of the Results part, not at the end of the paper.
Author Response
We thank the reviewer for his/her comments. We have attached our replies to the reviewer's report as a pdf file.

Reviewer 2 Report
The authors have presented a clay based coating approach for corrosion resistance. This is an interesting approach, the details aside that merits discussion. The authors should present some further insights into the background and viability of clay based coatings across diverse engineering materials. Why do we expect this mmt based coating to work?
The platelet structure and phylosilicate layers in the minerals should be examined and discussed as these are important in any clay based material. Describe the T-O-T layers, describe average number of layers in clay platelet.
Most importantly the coating and its robustness are not meaningfully characterised.
Is the coating bonded?
What holds to coating to the steel?
Coating adhesion measurement?
Scratch and wear resistance?
All of these are of paramount importance in the performance of the coatings examined here and yet are not presented or discussed.
Some improvement with the assistance of a native English speaker would make this more communicative and useful
Author Response
We thank the reviewer for his/her comments. We are attaching our replies to all the Comments and Suggestions for Authors as a pdf file.

Reviewer 3 Report
The paper relates to an essential problem of corrosion protection using an inhibitor based on nanosized montmorillonite clay. The obtained results indicate a significant reduction of corrosion intensity in HCl due to the application of this inhibitor which makes it promising for commercial implementation. However, the manuscript can be published only upon a significant revision on the following issues:
1. The brand and composition of the used steel, as well as the chemical composition of the applied montmorillonite should be indicated,
2. The brand of the balance and weighing precision should be indicated
3. Formulas for the calculation of all the considered characteristics (corrosion rate by both weight loss and polarization methods, inhibition efficiency, etc.) should be presented,
3. The weight loss and corrosion rates for blank samples in Tables 1 and 2 are calculated incorrectly, probably as well as all the relating inhibition efficiencies. As follows from these tables, the addition of relatively small MMT amounts results in a growth of the corrosion rate and inhibition is only observed at high MMT concentrations, which is probably erroneous.
4. In Tables 3 and 4. the meaning of such terms as “smoothing”, “calculation zone”, “segment” should be clarified. Furthermore, the valence indicated as 3, whereas in HCl iron exhibits the valence 2.
5. In Fig. 3, the description of the observed XRD peaks should be presented.
6. For SEM data, the corresponding MMT concentration should be indicated. The presence of carbon should be also discussed in more detail, particularly in respect to its content in MMT.
7. It is desirable to perform special tests on local corrosion (pitting) for the samples protected with the applied MMT because this kind of corrosion usually cannot be reliably detected by weight loss and polarization current experiments, while in the case on non-continuous coatings pitting is a very dangerous phenomenon.
8. The novelty of the presented results, at least a brief comparison of the considered data with those obtained by other researchers using similar materials should be discussed.
Author Response

(The authors gave the same response as above.)

Round 2
Reviewer 1 Report
The paper can be accepted in the present form
Author Response
We thank the reviewer for his/her comments and suggestions.
We highly appreciate the comment of the author :"The paper can be accepted in the present form".
Thank you again.
Reviewer 3 Report
The authors revised the manuscript in accordance with most of the comments. However, some issues still require clarification:
1. In Line 57 the abbreviation TOT should be disclosed
2. Every steel has its commercial brand, so it should be indicated to make the experiment description more clear.
3. In Tables 3,4 it seems better not to present the atomic mass, valence and density (furthermore, why it is indicated as 8 g/cm3 in Table 3 and 7.9 g/cm3 in Table 4?) but indicate these values in the experiment description or table heading.
4. In lines 175-176 it is better to present the data as 2q = 20.82o , etc.
5. As the main novelty of this article relates to the application of modified montmorillonite as a corrosion inhibitor, it would be desirable to compare the target inhibition performances with the application of raw montmorillonite clay as a reference experiment in addition to the inhibitor-free blank test.
6. The adhesion of MMT inhibitor layers to the steel surface should be characterized.
Author Response
Thank you for the comments.
Please find attached our replies to all your comments.
Hope this will satisfy.
Thanks again.
Authors
